# The Role of Health Institutions in Training Healthcare Personnel for the Digital Transition: The International Training Program of the Order of Physicians and Dentists of Rome

Maria Chiara Gatto [1,2,]*, Emanuela Maria Frisicale [3], Pietro Palopoli [4], Martina Sapienza [5], Emanuele Caroppo [6], Cristina Patrizi [7], Giovanni Migliano [7] and Gianfranco Damiani [5]

1   National Institute for Infectious Disease "Lazzaro Spallanzani" IRCCS, Via Portuense 292, 00149 Rome, Italy
2   Department of Cardiology, Sant'Eugenio Hospital, Piazzale dell'Umanesimo 10, 00144 Rome, Italy
3   General Direction of Health Prevention, Ministry of Health, Viale Giorgio Ribotta 5, 00144 Rome, Italy
4   Department of Mechanical and Aerospace Engineering, Polytechnic University of Turin, Corso Castelfidardo 39, 10129 Torino, Italy
5   Institute of Public Health, Section of Hygiene, Catholic University of the Sacred Heart, Via Garibaldi 28, 00153 Rome, Italy
6   Local Health Unit Roma 2, Viale Battista Bardanzellu 8, 00157 Rome, Italy
7   Order of Medical Surgeons and Dentists of Rome, Via Gubbio 36, 00181 Rome, Italy
*   Correspondence: mariachiaragatto@hotmail.com

**Abstract:** Digital health, encompassing the use of digital technologies in healthcare, and telemedicine, facilitating healthcare delivery across long distances, have witnessed widespread applications across various healthcare domains. The COVID-19 pandemic accelerated the adoption of digital solutions in healthcare, overcoming barriers to access and fostering transitions to new care models. However, healthcare professionals often lack digital health competencies, necessitating targeted training initiatives. This study presents a project initiated by the Order of Physicians and Dentists of Rome, promoting a comprehensive training program in digital health for healthcare professionals. This investigation aims to describe the project, report demographic characteristics of participants, and analyze survey results on participants' perceptions of the training program. The Erasmus+ project, titled 'Training of Physician Trainers in Telemedicine, eHealth, and Digital Medicine,' facilitated the digital transition of the healthcare sector through international training. The project involved structured courses, job-shadowing, and support activities in Malta and Madrid. A survey, developed using the Delphi methodology, assessed participants' views on telemedicine. Thirty participants, selected based on merit, engaged in the project. Survey responses highlighted a strong impact on participants' understanding of digital health concepts and increased confidence in utilizing digital tools. Notably, 85% acknowledged significant skill acquisition in healthcare digitalization. The project addressed a critical training gap among healthcare professionals, emphasizing the need for ongoing education in digital health. Despite existing recommendations, formal digital health education remains limited. The study underscores the importance of educational efforts to foster a digitalized healthcare model.

**Keywords:** digital health; telemedicine; healthcare professionals; training; Erasmus+ project

## 1. Introduction

Digital health is defined as the use of digital technologies for health and healthcare, while telemedicine represents the delivery of healthcare services where distance is a critical factor. In recent decades, these concepts have been applied to several branches of healthcare, from diagnosis to prevention, treatment, adherence, lifestyle, and patient engagement [1,2]. The use of digital solutions in medicine can decrease errors and costs, increase productivity and efficiency, and allow shared decision-making and self-advocacy for patients [3].

Furthermore, part of the digital health theme includes the digital health wallet, a digital platform that consolidates health-related information and has garnered significant

attention in recent years. In the literature, there are consensus documents regarding the use of health wallets, with particular interest in the topics of privacy and cybersecurity [4].

The COVID-19 pandemic, while causing health and economic crisis worldwide, favored the transition to digital solutions in many industrial fields and in society as a whole. Healthcare and education can be listed among the fields that benefited the most from digital transition [5].

Digital tools, such as apps for patient tracing, remote triage emergency services, and video-visits, have already been proven to be effective, including during the pandemic, when access to health services is restricted. However, many of the solutions that were introduced and implemented during the emergency have then been consolidated, contributing to the definition and adoption of new digital models of care. Nowadays, there are strong pushes for healthcare systems to be digitally enhanced and converted [6] and patients expect healthcare providers to offer digital tools as part of healthcare service delivery [7]. However, healthcare providers and students demonstrate a lack of digital health competencies and highlight the need for further training in digital health [8].

As a consequence, there is a compelling need for present and future clinicians to acquire digital health competencies and skills [9]. Medical schools worldwide have already started to introduce digital health education in their programs and studies have been published assessing digital health courses for medical students [10,11].

Yet, few providers receive digital-health-specific education during their academic careers, even though this training is encouraged by their respective organizations [12]. For instance, the American Medical Association and the National Organization of Nurse Practitioner Faculties (NONPF) recommend telehealth training of medical students, nurse practitioners, and residents and recognize that the use of telehealth is critical in reducing healthcare barriers [13,14].

However, telehealth training integrated into trainee education or healthcare provider education continues to be deficient [15]. Furthermore, despite the continuing advancement in telemedicine, historically, the clinical workflow has been developed for a face-to-face model of care. Moreover, senior graduates did not receive any education on digital health and digital technologies in medical school. For this category, postgraduate continued education would be of paramount importance. As a result, overall, the global transition to a digital model of health is far from adequate. To switch to a consolidated digital model of care, education is the field where efforts should be directed; healthcare organizations can certainly have an important role in the digital training of care providers.

The COVID-19 pandemic has accelerated the process of digitizing healthcare and transforming the modes of accessing health services. The use of remote services has highlighted the need for healthcare professionals to be adequately trained and prepared. The patient's 'home' must become the 'primary place of care.' This revolution is now possible thanks to 'telemedicine' and integrated home care.

In this regard, the Order of Physicians and Dentists of Rome promoted the training of thirty doctors and dentists through the Erasmus+ program.

To date, no formal research has been conducted on digital health education integration. The aims of the present investigation are as follows:

- To describe a project promoted by the "Order of Physicians and Dentists of Rome" and the development of an international training program on digital health;
- To report the demographic characteristics of the care providers involved;
- To report the results of a survey on the participants' perceptions of the training program.

## 2. Materials and Methods

The Erasmus+ project titled 'Training of Physician Trainers in Telemedicine, eHealth, and Digital Medicine, for the Digital Transition of the Health Care Sector,' promoted by the Order of Physicians and Dentists of Rome, enables the healthcare sector to accelerate its digital transition through the training of medical trainers in telemedicine, eHealth, and digital medicine. Its overall objective is to support the effort aimed at training teaching physicians

to promote a common strategy for global dedication to citizens' health. The training of healthcare personnel took place in a European context. The healthcare staff participated in a 10-day course titled "Digital Health for 21st Century Healthcare Professionals" and visited hospitals, universities, companies, and healthcare facilities where telemedicine practices were implemented. As a result of the project, participants were surveyed regarding the use of telemedicine.

*2.1. Survey Development and Delphi Assessment*

The survey was developed by medical doctors and dentists among the participants of the Erasmus+ group. The survey was conducted to obtain a subjective evaluation before and after the training experience. The training program included in-person classroom lectures and internships at hospitals and dental clinics in Madrid and Malta.

The classroom lectures were divided into five modules: (1) Innovation in Health: The Digital Transformation of the Health Sector; (2) Connected Professionals and Patients. Telehealth and Mobile Health; (3) Big Data in Health: Algorithms, Big Data, and Artificial Intelligence; (4) Emerging Technologies for the Present and Future; (5) Trends in Healthcare: New Digital Models. In order to validate the questionnaire, the Delphi methodology was pursued [16,17]. As a minimum number of experts is not defined [18], the OMCEO Roma and the working group proposed to send the questionnaire to some experts (medical doctor/professors) in digital health and telemedicine. The final survey was composed of 20 questions: a few questions were open ($n = 2$), some were multiple choice ($n = 4$), and others were based on a 10-point Likert scale ($n = 14$). the data derived from the numerical questionnaire were reprocessed as follows: 0–2: strongly disagree; 3–4: disagree; 5–6: neutral; 7–8: agree; and 9–10: strongly agree.

*2.2. Statistical Analyses*

Continuous variables were reported as mean and standard deviation (SD) or median and 25–75th Interquartile Range (IQR), when indicated, while categorical variables were reported as absolute number and percentage. Differences between groups were evaluated using the Student's *t*-test or Mann–Whitney test; the difference between proportions was calculated using chi-square or Fisher's exact tests, as appropriate. For data that did not have a normal distribution, the Wilcoxon matched-pairs signed-rank test was employed. The data were recorded in anonymized spreadsheets, and the statistical analysis was blindly performed using Prism GraphPad (v.8 produced by Dotmatics, Boston, MA, USA) or STATA (v.17 produced by StataCorp LLC, College Station, TX, USA) software, when appropriate.

**3. Results**

*3.1. Project Description*

The project promoted by the Order of Physicians and Dentists of Rome was titled 'Training of Physician Trainers in Telemedicine, eHealth, and Digital Medicine, for the Digital Transition of the Healthcare Sector' and was submitted by the organization under the Erasmus+ Program 2021–2027—Action KA1 Individual Mobility for Learning. The project envisioned the realization of 30 mobilities intended for medical personnel with the role of trainer and enrolled with the promoting organization.

The project involved structured courses with job-shadowing and support activities at host institutions in Malta and Madrid, with a focus on telemedicine, digital health, and the reform of the medical sector at European and global levels during the COVID-19 pandemic. Participation in the program included preparatory activities, the production of educational materials during and after the mobility experience, and documentation of the activities carried out.

Approximately 90 applications were collected from doctors with the following characteristics:

- Sufficiently autonomous knowledge of the English or Spanish languages.
- Intermediate-level computer skills.

- Willingness to disseminate the knowledge and skills acquired after mobility within their institution and among colleagues through project dissemination activities.
- Aged between 25 and 65 years.
- Proven experience in teaching activities.

In order of merit, 30 participants were selected (10 dentists and 20 physicians) for participation in the project (15 assigned to Malta and 15 assigned to Madrid). All participants were given the opportunity to participate anonymously in a survey on their perceptions of the training offer.

### 3.2. Demographic Characteristics of the Participants

Out of 30 project participants, 28 (93%) responded to the survey. Approximately 70% of the participants were female, and 68% were under the age of 40. General practitioners and dentists represented 23% and 33% of the participants, respectively, while specialists in various disciplines accounted for 43%.

### 3.3. Survey Results

A total of 86% of participants stated that there were no courses on telemedicine within the degree program they attended and only 14% of the participants stated that telemedicine had been previously addressed in an academic context through lectures, practical exercises, and webinars. However, 61% of the participants had addressed the topic of telemedicine in other postgraduate training courses, mostly through conferences or update courses (80%), while others pursued more structured educational paths such as master's programs and doctoral research (20%).

In 54% of cases, the participants expressed that utilizing digitization in medical-surgical training is a very useful method.

The subsequent survey questions aimed to analyze participants' perceptions of their skills in telemedicine and digital health before and after their participation in the project. The participants responded to the questions using a rating scale from 0 to 10, where 0 indicated complete disagreement and 10 represented the highest level of agreement.

The total score assigned by participants to each question, before and after the course, is reported in Table 1. In the same table, the maximum score obtained for each question before and after the course is also reported, along with their percentage difference.

**Table 1.** In this table, the questions and scores assigned by the participants before and after attending the course are summarized.

| N | Questions | Sum of Scores | | *p* Value | Maximum Scores | | Delta |
|---|---|---|---|---|---|---|---|
| | | Before Training | After Training | | Before Training | After Training | |
| 1 | How important do you think the concept of 'innovation' is? | 8.5 ± 1.6 | 9 ± 1.4 | 0.08 | 50% | 75% | 25% |
| 2 | How clear is the concept of 'digital health'? | 6.4 ± 2.5 | 8.4 ± 1.3 | 0.0001 | 21% | 43% | 22% |
| 3 | How do you assess the level of your 'digital literacy'? | 5.9 ± 2 | 7.8 ± 1.1 | 0.0001 | 7% | 21% | 14% |
| 4 | How confident would you feel about initiating and conducting telemedicine services in your work environment? | 5.8 ± 2.9 | 7.8 ± 2.1 | 0.0001 | 18% | 44% | 25% |
| 5 | How feasible do you consider providing telemedicine services in your operational context to be? | 5.8 ± 3 | 6.9 ± 2.7 | 0.0009 | 18% | 29% | 11% |

**Table 1.** *Cont.*

| N | Questions | Sum of Scores | | *p* Value | Maximum Scores | | Delta |
|---|---|---|---|---|---|---|---|
| | | Before Training | After Training | | Before Training | After Training | |
| 6 | How strongly would you recommend to a patient the use of apps (compliant with GDPR and supported by scientific evidence) to monitor their health status? | 6 ± 2.6 | 8.4 ± 1 | 0.0001 | 18% | 50% | 32% |
| 7 | How often do you use telemedicine applications? | 3.6 ± 2.8 | 4.9 ± 2.8 | 0.0001 | 0% | 11% | 11% |
| 8 | How strongly would you recommend to one of your patients the use of sensors that transmit vital parameters remotely? | 5.8 ± 2.9 | 8 ± 1.7 | 0.0001 | 14% | 46% | 32% |
| 9 | How useful do you consider the use of a chatbot for triaging outpatient patients to be? | 4 ± 2.8 | 6.3 ± 2.8 | 0.0001 | 7% | 22% | 15% |
| 10 | How useful do you consider the use of big data and artificial intelligence for scientific research purposes to be? | 6.5 ± 2.3 | 8.3 ± 1.4 | 0.0001 | 18% | 46% | 28% |
| 11 | How important do you consider the knowledge of methodologies for acquiring structured data to be? | 6.1 ± 2.6 | 8.1 ± 1.3 | 0.0001 | 18% | 32% | 14% |
| 12 | How useful do you consider the use of big data and artificial intelligence for healthcare assistance to be? | 6.8 ± 2.1 | 7.1 ± 2.3 | 0.0013 | 11% | 25% | 14% |
| 13 | How useful do you consider the use of big data and artificial intelligence for prevention to be? | 6.6 ± 2.3 | 8.5 ± 1.4 | 0.0001 | 25% | 50% | 25% |
| 14 | How sustainable (both socially and economically) do you consider the use of telemedicine to be? | 6.6 ± 2 | 8.4 ± 1.3 | 0.0001 | 14% | 42% | 29% |

In the first column, the reference numbers of the questions are listed, corresponding to the questions expressed in the second column. The third and fourth columns report the scores (mean and standard deviation) given by the participants before and after attending the course, respectively. The fifth column indicates the p value. In the sixth and seventh columns, the maximum scores assigned to each question before and after the course delivery are expressed. Finally, in the eighth column, the percentage difference between these maximum scores before and after the course is calculated.

Approximately 90% of the participants reported using technological tools to assess/manage patients remotely. Voice calls and instant messaging are used by 80% of the participants, followed by email at 75% and video calls at 57%. Only in 30% of cases did participants report using dedicated telemedicine platforms.

Participants were also asked about the specializations that could benefit the most from the use of telemedicine. The results are reported in Table 2.

**Table 2.** In this table, some of the specializations to which, according to the participants, telemedicine can be applied are listed.

| Specialties | Mean ± SD |
|---|---|
| Public Health and Hygiene | 8.6 ± 1.7 |
| General Medicine | 8.3 ± 2 |
| Cardiology | 8 ± 1.6 |
| Dermatology | 8 ± 1.3 |
| Radiology | 8 ± 1.3 |
| Emergency Medicine | 7.9 ± 2 |
| Psychiatry | 6.1 ± 2.3 |
| Surgery and Surgical Specialties | 6.1 ± 2.4 |
| Dentistry | 4.8 ± 2.2 |

In the first column, the specialties are listed; in the second column, the scores assigned on a scale from 0 to 10 are expressed as the mean ± standard deviation.

The importance of technological innovation was well known by half of the participants before the start of the course, and this awareness increased to 75% after the course itself.

## 4. Discussion

The concept of innovation is of paramount importance from the perspective of organizing training courses on the topics of digital health and telemedicine; but, as seen from the statistical analysis, there is no statistically significant difference regarding the perception of the concept of technological innovation. This can be explained by the fact that the awareness level among healthcare professionals was already very high before participating in the project. This objective data is certainly influenced by the selection bias of participants based on merit order and suggests that those selected for participation in the project already had a positive predisposition towards updates and technological innovation in the healthcare profession. Analyzing the survey responses globally, it is possible to identify some groups of questions related to behaviors and/or attitudes upon which the course seems to have had a greater impact (with a delta grading $\geq 25\%$). Indeed, at the end of the course, 44% of the participants stated that they feel very confident implementing telemedicine within their work environment and about half of them would strongly recommend the use of apps (50%) and sensors (44%) to monitor their patients in telemedicine. Regarding the use of artificial intelligence and big data, at the end of the course, 46% of the participants believed that these methodologies can be widely used for scientific research in healthcare, while 50% believed they can be equally used to promote preventive measures. Finally, about half of the participants believed that the use of telemedicine is a very feasible practice in terms of social and economic sustainability.

At the end of the survey, 85% of the participants stated that participation in the project significantly influenced their acquisition of specific skills in the field of healthcare digitalization.

The perception among trained dentists was that telemedicine is still not widely applicable in this field, although it could find great application in terms of tele-prevention. In this regard, there are virtual reality applications that teach proper oral hygiene maneuvers, although this presupposes the use of devices and software that could have high costs.

Telemedicine is configured within a healthcare system as a social and healthcare care methodology. In recent decades, with the evolution of technology, this methodology has also been incrementally and rapidly improved and is still in continuous evolution. Training on telemedicine should begin as methodological education during undergraduate courses, and continuous updates should also occur in the postgraduate period. The generation of specialized medical doctors currently practicing in Italy (aged between 25 and 65) is experiencing a training gap that needs to be addressed. Institutions and scientific societies frequently offer scientific updating courses on the topic of telemedicine. The Order of Physicians and Dentists of Rome, for the first time, promoted a training course on digital health aimed at educating healthcare professionals who, in turn, will be promoters of further training and the development of telemedicine projects in individual hospital settings. The project, organized in this way, promoted the training of physicians and dentists in an international European context. At the end of the international course, the participants had a very positive perception of the training offered, and residential training events have already been organized, promoting the 'training cascade,' which is the main project's objective.

## 5. Conclusions

This research focused on the crucial significance of digital health and telemedicine in transforming the landscape of healthcare provision, especially following the onset of the COVID-19 crisis. The rapid adoption of digital solutions during the crisis highlighted their effectiveness, prompting a lasting shift towards new digital care models. Our project, spearheaded by the Order of Physicians and Dentists of Rome, aimed to bridge the existing gap in digital health competencies among healthcare professionals. The Erasmus+ initiative

facilitated the training of medical trainers in telemedicine, eHealth, and digital medicine, offering a unique opportunity for participants to immerse themselves in these evolving fields. The survey results from the participants revealed a positive impact on their perceptions and skills related to telemedicine and digital health. Notably, a significant proportion expressed increased confidence in implementing telemedicine services, recommending digital tools to patients, and embracing technologies such as artificial intelligence for healthcare research and prevention. Despite these promising outcomes, the study underscores persistent gaps in the formal integration of digital health education into medical curricula. The need for ongoing education, during both undergraduate and postgraduate training, becomes evident, particularly as the current generation of healthcare professionals grapple with a training deficit in digital health. Our findings emphasize the importance of innovative training initiatives, such as the one described in this paper, in addressing the evolving needs of healthcare professionals. The positive feedback from participants underscores the potential of such programs to contribute to the digital transformation of healthcare. In conclusion, as we navigate the transition to a consolidated digital model of care, it is imperative that education remains a focal point. Healthcare organizations, academic institutions, and professional bodies play a crucial role in providing comprehensive digital health training. This study serves as a testament to the effectiveness of targeted programs in enhancing healthcare professionals' skills and perceptions in the ever-evolving landscape of digital health.

**Author Contributions:** Conceptualization, M.C.G.; methodology, E.M.F.; software, M.C.G. and E.M.F.; validation, E.C., C.P., G.M. and G.D.; formal analysis, M.C.G.; investigation, E.M.F., M.S. and P.P.; resources, G.M., C.P. and G.D.; data curation, M.C.G., E.M.F. and P.P.; writing—original draft preparation, M.C.G., E.M.F. and P.P.; writing—review and editing, M.C.G. and P.P.; visualization, E.C., C.P., G.M. and G.D.; supervision, C.P., G.M. and G.D.; project administration C.P., G.M. and G.D.; funding acquisition, C.P., G.M. and G.D. All authors have read and agreed to the published version of the manuscript.

**Funding:** This research received no external funding.

**Institutional Review Board Statement:** This study was conducted in accordance with the Declaration of Helsinki.

**Informed Consent Statement:** Informed consent was obtained from all subjects involved in the study. Since it is a survey, all participants were asked to provide voluntary consent for the processing of personal data and privacy. Approval from the local ethics committee is not applicable.

**Data Availability Statement:** It is possible to access the data by sending a request to the corresponding author.

**Acknowledgments:** We thank all colleagues who participated in the training program, sharing with all of us a unique and mutually enriching experience: Ombretta Papa, Gabriele Ordine, Giulia D'Amone, Henry Meza, Eleonora Mastria, Anissa Jaljaa, Carpico Elisa, Flavia Ciappina, Francesca Freda, Giorgia Carloni, ssa Giorgia Cunicella, Riccardo Giacomuzzi, Andrea Barbara, Menelaos Karpathiotakis, Fracassi Francesco, Angela Del Prete, Marco Scattaglia, Rosiello Francesco, Ferro Giorgia, Francesco Dojmi Di Delupis, Chiara De Angelis, Amedeo Coppola, Giulia Salustri, Marco Papa, and Peluso Laura.

**Conflicts of Interest:** The authors declare no conflicts of interest.

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
