# Peer review of "The Role of Health Institutions in Training Healthcare Personnel for the Digital Transition: The International Training Program of the Order of Physicians and Dentists of Rome"

_ime, doi:10.3390/ime3010008_

Round 1

Reviewer 1 Report

Comments and Suggestions for Authors

Thank you for the opportunity to review this interesting paper:

The article emphasizes the positive impact of digital solutions on reducing errors, increasing efficiency, and facilitating shared decision-making for patients.

1.     Please state European rules and directives about using digital health

You can find useful article here:

To Wallet or Not to Wallet: The Debate over Digital Health Information Storage

https://www.mdpi.com/2073-431X/12/6/114

2.     Please state for Informed Consent Statement: “Informed consent was obtained from all subjects involved in the study

In which way is it obtained, in written form? And how do you keep and protect data?

3.     I found some sentences generated with chat GPT,as

„This study sheds light on the pivotal role of digital health and telemedicine in reshap- 242

ing healthcare delivery, particularly in the wake of the COVID-19 pandemic.

Please  change it.

4.     I could not find an ethical approval for this research. It is necessary for processing the article and please state clearly.

Comments on the Quality of English Language

please remove chat GPT-generated text

Author Response

Dear Reviewer, thank You for your comments. We provide a point to point response:

  1. Please state European rules and directives about using digital health. You can find useful article here:  To Wallet or Not to Wallet: The Debate over Digital Health Information Storage https://www.mdpi.com/2073-431X/12/6/114

1. Thank you for the comment. I have read the article you mentioned with interest and it has been mentioned in the introduction and added to the bibliography.

  1. Please state for Informed Consent Statement: “Informed consent was obtained from all subjects involved in the study. In which way is it obtained, in written form? And how do you keep and protect data?

2. Thank you for the comment. The informed consent of all participants has been obtained in written form. This information has been added to the paper.

  1. I found some sentences generated with chat GPT,as „This study sheds light on the pivotal role of digital health and telemedicine in reshaping healthcare delivery, particularly in the wake of the COVID-19 pandemic. Please  change it.

4. Thank you for the comment. The sentence has been modified to “This research focus on the crucial significance of digital health and telemedicine in transforming the landscape of healthcare provision, especially following the onset of the COVID-19 crisis.”

  1. I could not find an ethical approval for this research. It is necessary for processing the article and please state clearly.

4. Thank you for the comment. Since it is a survey, all participants were asked to provide voluntary consent for the processing of personal data and privacy. Additionally, no sensitive data were collected, and the survey was conducted anonymously by all participants. In this case, approval from the ethics committee is not applicable.

- Comments on the Quality of English Language. please remove chat GPT-generated text

- Thank you for the feedback, the text has been modified.

Reviewer 2 Report

Comments and Suggestions for Authors

Interesting article that is relevant in the post pandemic period.  The adoption of technology made it efficient to administer healthcare during the pandemic and much has continued even in the post pandemic period. I was surprised about the knowledge gap in adapting to the newer technology.  Nevertheless, this article identifies and opportunity to enhance the education and during the early training years.  

Comments on the Quality of English Language

This is a well written paper from the grammar and language standpoint.  

Author Response

Thank you for the comment. Indeed, there is a noticeable gap in knowledge and skills regarding telemedicine, which in some cases may be attributed to the age of the participants. I believe that the publication of this article provides a snapshot of the current landscape, and perhaps in the near future, the same survey may demonstrate a homogenization of knowledge among all participants.

Reviewer 3 Report

Comments and Suggestions for Authors

This manuscript describes a training program on digital health and reports the demographics of the medical professionals who participated in this program as well as their evaluation of the program. Given the rise of digitalization during the COVID-19 pandemic, I think assessment of programs like this are of vital interest to the research community. I also thought that the methodology and analysis of the results of this study were presented in straightforward and easy-to-understand manner, although I do think the authors need to make it more clear that was a pre-/post-study design, in which participants completed the same survey before and after the program. Thus, I recommend that the manuscript be accepted after minor revisions, which I will describe below.

1). Because one of the state purposes of the manuscript was to describe this digital health education program, I would like to see more information about the content and specific activities that participant did during this 10-day course. Specifically, how were the activities related to telemedicine and digital health?

2) I'd also recommend that the authors provide a little more information about the survey development, rather than putting it all in the supplemental materials. For example, how many were open-ended questions, multiple-choice questions, or Likert-scale questions? Of the questions presented in Table 1, were they all designed to assess the same theme/topic or various themes?

3) I thought an interesting point in this manuscript was that the program included both dentists and physicians as participants. Although statistical power is probably lacking due to the small sample size, I'd like to know of the authors found any differences in these two groups' perceptions of the training program. For example, based on the results in Table 2, it seems that participants felt that Dentistry would benefit much less from telemedicine than some of the medically-related fields. Perhaps the dentists' evaluation of this program may be slightly lower than the physicians, because they think telemedicine is less applicable for their field?

Comments on the Quality of English Language

There were a few sentences that seemed to be grammatically incorrect, such as lines 58-59, and "graduated" in line 74 (I think it should be "graduates"). There were also some unusual word choices, in my opinion, such as "previewed questions" in line 116, and "reprocessed" in line 118. I recommend that the authors double-check their English again 

Author Response

Dera Reviewer, thank You for the comment. We provide a poin to point response:

R: This manuscript describes a training program on digital health and reports the demographics of the medical professionals who participated in this program as well as their evaluation of the program. Given the rise of digitalization during the COVID-19 pandemic, I think assessment of programs like this are of vital interest to the research community. I also thought that the methodology and analysis of the results of this study were presented in straightforward and easy-to-understand manner, although I do think the authors need to make it more clear that was a pre-/post-study design, in which participants completed the same survey before and after the program. Thus, I recommend that the manuscript be accepted after minor revisions, which I will describe below.

A: Thank you for the comment. In the materials and methods section, we specified that it is a subjective evaluation conducted before and after the training experience.

1). Because one of the state purposes of the manuscript was to describe this digital health education program, I would like to see more information about the content and specific activities that participant did during this 10-day course. Specifically, how were the activities related to telemedicine and digital health?

1) Thank you for the comment. In the methods section, the training program has been included, which involved both classroom lectures and internships at hospitals and dental centers in Madrid and Malta.

2) I'd also recommend that the authors provide a little more information about the survey development, rather than putting it all in the supplemental materials. For example, how many were open-ended questions, multiple-choice questions, or Likert-scale questions? Of the questions presented in Table 1, were they all designed to assess the same theme/topic or various themes?

2) Thank you for the comment. Most of the questions (14) were conducted using the Likert scale, while only a few questions (4) required multiple-choice answers. Only two questions question regarding any additional comments was open-ended.

3) I thought an interesting point in this manuscript was that the program included both dentists and physicians as participants. Although statistical power is probably lacking due to the small sample size, I'd like to know of the authors found any differences in these two groups' perceptions of the training program. For example, based on the results in Table 2, it seems that participants felt that Dentistry would benefit much less from telemedicine than some of the medically-related fields. Perhaps the dentists' evaluation of this program may be slightly lower than the physicians, because they think telemedicine is less applicable for their field?

3) Thank you for the comment. You correctly deduced the perception of the training offering by dentists: the perception among trained dentists was that telemedicine is still not widely applicable in this field, although it could find ample space in terms of tele-prevention. In this regard, there are virtual reality applications that teach proper oral hygiene maneuvers, although this presupposes the use of devices and software that could have high costs.  We have incorporated these observations into the discussion section.

Comments on the Quality of English Language: There were a few sentences that seemed to be grammatically incorrect, such as lines 58-59, and "graduated" in line 74 (I think it should be "graduates"). There were also some unusual word choices, in my opinion, such as "previewed questions" in line 116, and "reprocessed" in line 118. I recommend that the authors double-check their English again 

A:Thank you for the comment, the text has been revised.